# Identification, Expression and Evolutional Analysis of Two *cyp19-like* Genes in Amphioxus

**DOI:** 10.3390/ani14081140

**Published:** 2024-04-09

**Authors:** Yajun Wang, Jingyuan Lin, Wenjin Li, Guangdong Ji, Zhenhui Liu

**Affiliations:** College of Marine Life Sciences, Institute of Evolution & Marine Biodiversity, Ocean University of China, Qingdao 266003, China; wangyajun419@163.com (Y.W.); linjingyuan2023@163.com (J.L.); lwj08042@126.com (W.L.); jamesdong@ouc.edu.cn (G.J.)

**Keywords:** amphioxus, invertebrates, sex differentiation, *cyp19*

## Abstract

**Simple Summary:**

The mechanism of sex determination and differentiation in animals has remained a central focus of reproductive and developmental biology research. Among these, the study of sex differentiation mechanisms in amphioxus, an ancient and unique organism, holds significant value for understanding the evolution and origin of sex determination mechanisms in vertebrates. However, the current understanding of the sex differentiation regulatory mechanisms in amphioxus remains limited. This study aims to enrich and deepen our comprehension of the sex differentiation mechanisms in amphioxus through the identification of two *cyp19* homologous genes and the analysis of their expression level in male and female gonads. Additionally, this research provides valuable insights into the formation and evolutionary pathways of sex determination mechanisms in vertebrates.

**Abstract:**

The mechanism of sex determination and differentiation in animals remains a central focus of reproductive and developmental biology research, and the regulation of sex differentiation in amphioxus remains poorly understood. Cytochrome P450 Family 19 Subfamily A member 1 (*CYP19A1*) is a crucial sex differentiation gene that catalyzes the conversion of androgens into estrogens. In this study, we identified two aromatase-like genes in amphioxus: *cyp19-like1* and *cyp19-like2*. The *cyp19-like1* is more primitive and may represent the ancestral form of *cyp19* in zebrafish and other vertebrates, while the *cyp19-like2* is likely the result of gene duplication within amphioxus. To gain further insights into the expression level of these two *aromatase-like*, we examined their expression in different tissues and during different stages of gonad development. While the expression level of the two genes differs in tissues, both are highly expressed in the gonad primordium and are primarily localized to microsomal membrane systems. However, as development proceeds, their expression level decreases significantly. This study enhances our understanding of sex differentiation mechanisms in amphioxus and provides valuable insights into the formation and evolution of sex determination mechanisms in vertebrates.

## 1. Introduction

Sex determination and differentiation, crucial for species continuation, is a complex process influenced by various factors within and outside cells. It involves genetic gonad differentiation shaped by the environment, with multiple cells and organs participating. Gender differentiation builds on this to determine an organism’s female or male phenotype. Despite diverse sex determination mechanisms among animals, they all rely on the products of gene expression. Therefore, delving into these molecular mechanisms holds profound significance. Vertebrates show diverse and uncertain sex determination mechanisms; yet, conserved pathways and genes exist across species. For instance, *Sry* is a switch gene in mammals [1], and other genes related to female and male sex formation have been discovered, such as *CYP19A1*, *WNT4*, *FOXL2*, *AMH*, *SF1*, *DMRT1,* and *DAX1* [2,3,4,5,6]. Signaling molecules like BMP and WNT also regulate sex differentiation [7,8,9,10]. In invertebrates, there exist a wide range of sex determination mechanisms. Primary sex determination signals often exhibit variations even among closely related species and can rely on a wide range of mechanisms, including genetic factors, environmental cues, or a combination of both [11]. In the crustacean *Daphnia magna*, for instance, a reduced photoperiod, lack of food, and high population density favor the production of males [12]. Also, sex determination can occur at different stages of an animal’s life cycle, ranging from fertilization to later stages in life where sex reversal may take place in sequential hermaphrodites [11].

As a Protochordata, the amphioxus bridges invertebrates and vertebrates, playing a key role in evolution [13]. Since its discovery in 1774, it has served as a valuable model for studying the origins of vertebrates [14,15]. With fewer lineage-specific changes than *Urochordate* [16], the amphioxus offers a simpler body structure for studying vertebrate origins and invertebrate evolution [17,18]. It exhibits sexual dimorphism with a 1:1 sex ratio [19,20]. The gonads, attached to the inner walls of the peribranchial cavity, are composed of 26 pairs of rectangular sacs with thick walls and its gonads can be clearly observed due to skin transparency.

Currently, there have been some studies attempting to explore the regulation of gonad development and the sex determination mechanism in amphioxus. Zhang et al. [21,22] demonstrated that estrogens and androgens play key roles in the maturation of amphioxus gonads. Fang’s research team [21,22] further elucidated the regulatory functions of 17β-estradiol, testosterone, progesterone, and other hormones in yolk formation and spermatogenesis. Mizuta and Kubokawa [23,24] cloned genes such as *cyp11a*, *cyp17*, and *cyp19* from amphioxus, providing insights into the biochemical pathways of steroid hormone synthesis. Additionally, although luteinizing hormone (LH) and follicle-stimulating hormone (FSH) have not been identified in amphioxus [25], thyrostimulin, a glycoprotein hormone, is considered a primitive precursor of these hormones [26,27,28]. Shi et al. [29] and Zhen et al. [30] revealed through genetic analysis that amphioxus possesses a ZW-type sex chromosome system. However, the sex determination mechanism in amphioxus remains largely unknown.

The CYP superfamily is extensive, with over 13,000 genes across 400 gene families [31]. The Cytochrome P450 Family 19 Subfamily A member 1 (*CYP19A1*) is crucial for sex differentiation. This enzyme converts androgens to estrogens [32], regulating vertebrate sex development [33,34,35,36]. In most vertebrates, there is only one aromatase gene. However, pigs have three aromatase genes, and the model fish zebrafish have two aromatase genes. In model fish zebrafish, there are two types of aromatase: ovarian and brain aromatase, encoded by the genes *cyp19a1a* and *cyp19a1b*, respectively. The ovarian aromatase plays a crucial role in sex differentiation. The *cyp19a1a* is primarily expressed in the ovaries, and its knockout in zebrafish leads to female-to-male sex reversal and delayed testis development [37,38,39,40,41]. The *cyp19a1b* is primarily expressed in the brain and has significant impacts on brain development and function. It is involved in neuroendocrine metabolism during gonad development and also maintains the testis [42,43]. Cyp19a1a and Cyp19a1b differ structurally and functionally, possibly due to genome duplication under selective pressure during evolution [42,44]. Over the course of evolution, vertebrate *cyp19a1* has exhibited high sequence conservation and good synteny [45].

Mizuta cloned amphioxus cytochrome P450 members, which encode proteins involved in catalyzing key reactions in the synthesis of progesterone, androgen, and estrogen from cholesterol, such as *cyp11a*, *cyp17*, and *cyp19* [23,24]. We searched the amphioxus (*Branchiostoma floridae*) genome and found two *cyp19-like* genes (*cyp19-like1* GenBank accession number: XP_035672840.1; *cyp19-like2* GenBank accession number: XP_035669280.1) on separate chromosomes. These genes do not cluster with zebrafish *cyp19a1a* or *cyp19a1b* but sit at the evolutionary tree’s base. In this paper, we cloned the two *cyp19-like* genes from amphioxus *B. japonicum* and used bioinformatics and molecular techniques to understand their structure and evolution. We also studied their expression in amphioxus tissues using various methods. This comprehensive approach allowed us to determine the expression level of *cyp19-like* genes in the male and female gonads of amphioxus, providing valuable insights into their roles in gonadal differentiation and development. 

## 2. Materials and Methods

### 2.1. Experimental Materials

Qingdao amphioxus (*B. japonicum*) were collected near the Shazikou sea area of Qingdao in China in early May. Immediately after collection, preliminary processing, such as cleaning, weighing, and recording, was performed. When sexually mature, amphioxus can be distinguished by the color of its gonads: the testes appear white, while the ovaries appear pale yellow. The HEK 293T human embryonic kidney cell line was purchased from the American Type Culture Collection (ATCC). Six-week-old ICR female mice were used for antibody preparation. During the process, we strictly adhere to the experimental animal handling guidelines to ensure reliable experimental results and animal welfare.

### 2.2. RNA Extraction and cDNA Synthesis

All steps were performed in an RNase-free environment, adhering to laboratory safety protocols. After being ground thoroughly with liquid nitrogen, amphioxus was mixed uniformly with Trizol reagent and its total RNA was extracted using the Total RNA Kit I (OMEGA, #R6834-01, Norcross, GA, USA). To ensure the integrity and purity of the RNA, we utilized a Neodrop fluorophotometer to measure its absorbance at 280 nm, 320 nm, 230 nm, and 260 nm. Furthermore, we performed agarose gel electrophoresis for additional verification. For the synthesis of amphioxus cDNA, the Evo M-MLV RT Mix Kit with Gdna Clean for qPCR (AG, AG11728, Accurate Biology, Changsha, China) was utilized. The genomic DNA was removed, followed by a precise reaction protocol—a 15 min incubation at 37 °C, followed by a brief 5 s incubation at 85 °C—within a PCR machine for reverse transcription, yielding the desired reaction solution. 

### 2.3. Cloning of the cyp19

Initially, a comprehensive BLAST search was conducted on the NCBI (ncbi.nlm.nih.gov, accessed on 23 June 2023), utilizing the zebrafish *cyp19a1a* sequence as a query (Appendix A) to explore the Florida amphioxus (*B. floridae*) database. Through this meticulous investigation, we identified two *aromatase-like* gene sequences (XP_035672840.1 and XP_035669280.1). Subsequently, by aligning the transcriptome and genome data of Qingdao amphioxus, we determined two homologous sequences, referred to as *cyp19-like1* (corresponding to XP_035672840.1) and *cyp19-like2* (corresponding to XP_035669280.1). Then, specific primers (Table 1) were designed to clone the open reading frame (ORF) sequences of *cyp19-like1* and *cyp19-like2* using PCR. The PCR reaction was performed in a 20 μL volume, containing 10 μL 2×Taq Master Mix, 200 ng cDNA, and 0.5 μL of each primer. The PCR conditions were as follows: 94 °C for 5 min; 94 °C for 30 s, 55 °C for 30 s, and 72 °C for 90s for 36 cycles; and extended at 72 °C for 5 min.

### 2.4. Bioinformatics Analysis

The exon and intron sequences of amphioxus *cyp19* were extracted using TBtools, a versatile bioinformatics toolbox. Then, these sequences were uploaded to the GSDS2.0 (Gene Structure Display Server 2.0, gao-lab.org, accessed on 29 December 2023) to generate illustrative gene structure diagrams. Subsequently, we aligned the amino acid sequences of the amphioxus Cyp19-like1 and Cyp19-like2 proteins with those of the zebrafish Cyp19a1a and Cyp19a1b proteins retrieved from NCBI using Megalign 7.1.0 to assess sequence similarity. Additionally, we employed SMART 7.1.0 (SMART: Main page (embl.de)) and PHYRE2.0 (PHYRE2) to predict signal peptides, transmembrane regions, functional domains, and three-dimensional structures of the Cyp19 proteins. Lastly, we gathered amino acid sequences for Cyp19 and steroid biosynthesis-related Cyp subfamilies (Cyp3a, Cyp17a, and Cyp21) from various species. These sequences were aligned using Clustal W in MEGA 10.1.6 software and a phylogenetic tree was constructed employing the maximum likelihood method to reveal the evolutionary history of the Cyp19.

### 2.5. Real-Time PCR 

Gene expression levels were quantified using ChamQ SYBR Color qPCR Master Mix (Vazyme, #Q431-02, Nanjing, China) in an ABI 7500 Fast Real-Time PCR System (Thermo Fisher Scientific, 81 Wyman Street, Waltham, MA, USA). The relative quantitative real-time RT-PCR was performed in a 20 μL volume containing 10 μL 2×ChamQ SYBR Color qPCR Master Mix, 200 ng cDNA, and 0.5 μL of each primer. The relative quantitative real-time RT-PCR conditions were as follows: 95 °C for 15 s; 95 °C for 5 s, 60 °C for 15 s, and 72 °C for 35 s for 40 cycles; and 95 °C for 15 s, 60 °C for 60 s, and 72 °C for 15 s. The expression of amphioxus *ef1α* [46] was utilized as an internal control. Relative expression changes were calculated using the 2^−ΔΔCt^ method. All real-time PCR experiments were conducted in triplicate and replicated three times to ensure consistency and reliability of the results. The primers utilized for these experiments are listed in Table 1.

### 2.6. Paraffin Section In Situ Hybridization (ISH)

Amphioxus was subjected to a 24 h starvation treatment, then divided into head, middle, and tail sections using sterilized scissors and tweezers. The tissues were fixed with 4% paraformaldehyde at 4 °C for 12 h. Afterwards, the processed tissues were embedded into melted wax blocks. The wax blocks containing the tissues were sliced into sections (8 μm) using a paraffin microtome. Concurrently, primer pairs P11/P12 and P13/P14 were utilized for cloning *cyp19-like1* and *cyp19-like2*, respectively (Table 1). The fragments were amplified to prepare probes for ISH, synthesizing sense or antisense probes using T7 RNA Polymerase and SP6 RNA Polymerase, respectively. The denatured probes were applied to the sections and incubated at 50 °C, allowing hybridization between the probes and the target mRNAs. Following hybridization, any unbound probes were removed. They were incubated with anti-digoxin antibody at 4 °C overnight; then, NBT/BCIP staining solution was added for color development at 28 °C. Finally, they were observed and images captured under a Carl Zeiss Microlmaging GmbH 37081 (Serien-Nr: 3517001605 Zeiss Axio Imager A1, Oberkochen, Germany).

### 2.7. Hematoxylin and Eosin Staining

Xylene was used to remove the paraffin from the amphioxus paraffin sections (5 μm) and then the sections were hydrated through a graded series of alcohol concentrations. Next, the nuclear DNA was stained blue by immersing the sections in hematoxylin solution for 1 min and the cytoplasm was stained red using eosin solution for less than 5 s. The sections were treated with xylene for 1 min, the slides sealed with neutral balsam, and, finally, the stained sections observed under a Carl Zeiss Microlmaging GmbH 37081 (Serien-Nr: 3517001605 Zeiss Axio Imager A1, Oberkochen, Germany).

### 2.8. Protocol for Generating Mouse Polyclonal Antibodies

The polyclonal antibodies used in this study were homemade by our laboratory. The immunogenic peptide sequences for Cyp19-like1 and Cyp19-like2 are N′-CREELKTAPPSDKPD-C′ and N′-CPSRDHKSLDVSRNL-C′, respectively. The KLH-conjugated peptides were synthesized by GenScript Corporation in Jiangsu Province, China. Mice were immunized via intramuscular injection in the hind legs, followed by a booster immunization 21 days later. The experimental group of mice were injected with immunogenic peptides, adjuvant (KX0210041), and PBS, while the control group of mice was injected with only adjuvant and PBS. Mouse sera were collected and antibodies were purified.

### 2.9. Western Blot

Mature female amphioxus were taken after fasting for 24 h, ground into powder in liquid nitrogen, and then lysis buffer was added to lyse on ice for 20 min. After centrifuging to separate cell debris, the supernatant was collected as total protein. The total protein was mixed with Loading Buffer and boiled for 7 min to denature it. They were loaded onto an SDS-PAGE gel for electrophoresis. After electrophoresis, the proteins were transferred to a PVDF membrane. After blocking the membrane for 1 h, the primary antibody (1:800) was incubated at 4 °C overnight. Then, the secondary antibody (1:4000) was incubated at room temperature for 1 h, followed by washing with PBST 5 times. Finally, the ECL working solution was prepared in the dark, added dropwise to the membrane, and quickly transferred to a chemiluminescence imager for exposure and imaging. The primary antibodies utilized in this study are anti-Cyp19-like1 and anti-Cyp19-like2 polyclonal antibodies homemade by our laboratory. The second antibody is Goat Anti-Mouse IgG H&L (HRP) (ab6789).

### 2.10. Immunohistochemical Staining

Mouse anti-Cyp19-like1 and anti-Cyp19-like2 polyclonal antibodies were homemade by our laboratory as described above. The amphioxus tissue sections (5 μm) were also prepared as above. After treating the tissue sections in a hybridization oven at 60 °C for 45 min, they were dewaxed and rehydrated. Then, they were treated with 3% hydrogen peroxide at room temperature for 15 min to quench the endogenous peroxidase activity. The sections were incubated with mouse anti-Cyp19-like1 or anti-Cyp19-like2 polyclonal antibodies overnight at 4 °C. The control sections were incubated with nonimmunized serum. After washing, they were incubated with the second antibody (1:800) at room temperature for 1 h. After the addition of 0.015% DAB (*w*/*v*), they were maintained in the dark for 5 min for the chromogenic reaction. They were observed and photographed under a Leica Microsystems CMS GmbH (DMI3000, Wetzlar, Germany).

### 2.11. Subcelluar Localization

Following the manufacturer’s standard protocol, HEK-293T cells were co-transfected with pcDNA3.1-Cyp19-like-eGFP and pcDNA3.1-eGFP plasmids using Lipofectamine 2000 (Thermo Fisher Scientific, 81 Wyman Street, Waltham, MA, USA). At 36 h post-transfection, the cells were fixed with 4% paraformaldehyde (PFA). Subsequently, the cells were washed three times with phosphate-buffered saline (PBS) and stained with DAPI for nuclear visualization. Using a Leica Sp8 confocal microscope (Wetzlar, Germany), high-resolution images of the transfected cells were captured.

### 2.12. Statistical Analysis

Statistical analysis was performed using GraphPad Prism 9 (San Diego, CA, USA, www.graphpad.com, accessed on 29 December 2023), with all assays conducted in triplicate technical and three biologic replicates. Data were analyzed using one-way ANOVA or two-tailed Student’s *t*-test and presented as means ± SD. Significance levels were set at * *p* < 0.05, ** *p* < 0.01, and *** *p* < 0.001, with ‘ns’ indicating non-significance.

## 3. Results

### 3.1. Cloning of the cyp19

Using female amphioxus cDNA as a template, the complete open reading frame (ORF) of the *cyp19-like* gene in amphioxus *B. japonicum* was cloned. The *cyp19-like1* has a total length of 1491bp, encoding 496 amino acids (Figure 1A), while the *cyp19-like2* has a total length of 1482bp, encoding 494 amino acids (Figure 1B). Sequencing results confirmed the accuracy of the sequences.

### 3.2. Bioinformatic Analysis of the cyp19

#### 3.2.1. Comparison of *cyp19* Structures between Zebrafish and Amphioxus

The zebrafish *cyp19a1a* gene has nine exons located on chromosome 18, while *cyp19a1b* has nine exons on chromosome 25. In amphioxus, *cyp19-like1* has 10 exons on chromosome 4 and *cyp19-like2* has 9 exons on chromosome 3 (Figure 2). Since the zebrafish *cyp19a1a* and *cyp19a1b* genes are located on different chromosomes and have different functions, we hypothesize that the two *cyp19-like* in amphioxus, located on different chromosomes, may also exhibit functional differences.

#### 3.2.2. Comparison of Sequence Similarity between Zebrafish and Amphioxus Cyp19

The amino acid sequence similarity between zebrafish *cyp19a1a* and *cyp19a1b* and amphioxus *cyp19-like1* and *cyp19-like2* is low (Figure 3A). Sequence alignment revealed that the similarity between the two amphioxus *cyp19-like* sequences is only 39.4%. The sequence similarity between *cyp19-like1* and *cyp19a1a* and *cyp19a1b* is 35.2% and 35.5%, respectively. The sequence similarity between *cyp19-like2* and *cyp19a1a* and *cyp19a1b* is 37.1% and 37.8%, respectively (Figure 3B).

#### 3.2.3. Prediction and Comparison of Cyp19 Structures between Zebrafish and Amphioxus

SMART website was used to predict signal peptides, transmembrane regions, and domains of Cyp19 in zebrafish and amphioxus. The results showed that both amphioxus and zebrafish possess a conserved cytochrome p450 domain. Zebrafish Cyp19a1a does not have a transmembrane region, while Cyp19a1b, Cyp19-like1, and Cyp19-like2 have a transmembrane region at the N-terminus (Figure 4A). Three-dimensional structure prediction of Cyp19 encoded by zebrafish and amphioxus revealed a high degree of similarity in their three-dimensional structures (Figure 4B), indicating potential similar functions.

#### 3.2.4. Synteny and Evolutionary Analysis of Cyp19

Through Cyp19 synteny analysis, it was found that the protein has synteny in humans, mice, zebrafish, and amphioxus. *Cyp19-like1* and *cyp19-like2* exhibit synteny, but *cyp19-like2* has stronger synteny with humans, mice, and zebrafish, suggesting it may play a more important role in sex determination (Figure 5A). The main function of Cyp19 is to participate in steroid synthesis. Among the Cyp superfamily, other subfamilies related to steroid biosynthesis include Cyp3a, Cyp17a, and Cyp21. The results showed that the three amphioxus Cyp19-like sequences belong to the Cyp19 subfamily. Amphioxus Cyp19-like1 clusters with invertebrates such as hydroids and sea lilies, indicating a more primitive evolutionary status. In contrast, Cyp19-like2 clusters with vertebrates (Figure 5B). 

### 3.3. Expression Level of Two cyp19-like Genes in the Gonads of Male and Female Amphioxus

#### 3.3.1. Differential Tissue Expression of *cyp19*

We have demonstrated through semi-quantitative, real-time PCR, and ISH techniques that there are differences in the expression of *cyp19-like1* and *cyp19-like2* in different tissues of adult amphioxus. Semi-quantitative and real-time PCR results show that *cyp19-like1* is expressed the highest in the gills, followed by the ovaries in amphioxus. *Cyp19-like2* is highly expressed in the ovaries but is also present in tissues such as the gills (Figure 6A,B). To further investigate the expression of *cyp19-like* in the gills of male and female amphioxus, we conducted real-time PCR using male and female gill templates. The results showed no significant difference in the expression of *cyp19-like1* and *cyp19-like2* in the gills (Figure 6C). ISH results show that both *cyp19-like1* and *cyp19-like2* have significant positive signals in the gills and ovaries (Figure 6D). The real-time PCR and ISH results are similar to the characteristic of high expression of *cyp19a1* in the ovaries of vertebrates. However, unlike vertebrates, *cyp19-like* are also highly expressed in the gills of amphioxus. 

#### 3.3.2. Expression of *cyp19* at Different Stages

During our laboratory breeding period, we observed the gonadal development of amphioxus and found that the gonads begin to develop in April, with the appearance of small black particles (gonadal primordia) on both sides of the amphioxus body. The development time of male and female gonads and both sides of the gonads is relatively consistent. In May, the gonads begin to mature and, at this time, they are transparent vesicular structures, and the black particles become larger and lighter with development. June to July is the peak period of maturity, with large and full gonads, rectangular shape, white testes in males, and yellow ovaries in females. The spawning peak occurs in July.

Based on observations of gonadal morphology and histology, we divided the gonadal development cycle of amphioxus into four stages: undifferentiated gonad (T0), gonadal primordium (T1), gonadal development (T2), and gonadal maturity (T3) (Figure 7A). We designate the undifferentiated gonads as being in the T0 period, which begins after ovulation or spermiation. The emergence of gonadal primordium marks the start of the T1 period. The initiation of specialization of primordial gonad cells into testes or ovaries signifies the commencement of the T2 period. Once fully mature testes or ovaries are formed, it is considered the T3 period. We used real-time PCR to detect the expression level. We used real-time PCR to detect the expression level of the two *cyp19-like* at different stages of gonadal development in amphioxus. The results showed that both *cyp19-like1* and *cyp19-like2* are highly expressed at the beginning of gonadal development, i.e., the gonadal primordium stage, and their expression decreases with the maturation of the gonads (Figure 7B).

#### 3.3.3. Temporal Expression and Cellular Localization of Cyp19 

The immunogenic peptide sequences of Cyp19-like1 and Cyp19-like2 are N′-CREELKTAPPSDKPD-C′ and N′-CPSRDHKSLDVSRNL-C′, respectively. These immunogenic peptide sequences are located on the outer side of the predicted Cyp19 protein three-dimensional structure (Figure 8A). The antibody preparation effect was enhanced by KLH coupling. We used WB to detect the specificity of the prepared Cyp19 polyclonal antibody and the size of the Cyp19. The results showed good specificity of the Cyp19 antibody (Figure 8B).

We selected amphioxus individuals at three different stages and detected the protein expression levels of the two Cyp19-like at different stages of gonadal development using immunohistochemical staining. The immunohistochemical staining results were consistent with the real-time PCR results, showing that Cyp19-like1 and Cyp19-like2 are highly expressed at the beginning of gonadal development, i.e., the gonadal primordium stage, and their expression gradually decreases with the maturation of the gonads. Additionally, the expression level of Cyp19-like2 was higher (Figure 8C). Furthermore, we examined the localization of Cyp19 in cells and found that the protein is a membrane protein mainly localized to mitochondrial membranes, endoplasmic reticulum membranes, and other microsomal membrane systems (Figure 8D).

## 4. Discussion

Aromatase, encoded by the *cyp19a1*, plays a crucial role in vertebrate reproduction by catalyzing the conversion of androgens to estrogens. While aromatase is known to be critical in sex determination and differentiation in animals, the origin of the *cyp19a1* remains enigmatic. In vertebrates, the majority of animals possess a single aromatase gene. However, fish belonging to the model fish zebrafish exhibit a unique feature: they possess two aromatase genes, ovarian aromatase and brain aromatase. These genes are expressed in distinct locations and exhibit structural and functional differences. Through sequence alignment, we discovered the existence of two *aromatase-like* in Florida amphioxus, situated on chromosomes 4 and 3, respectively. Although the *cyp19a1a* and *cyp19a1b* genes in zebrafish reside on separate chromosomes and exhibit distinct functionalities, our understanding of whether the two *cyp19* homologous genes in amphioxus, which occupy different chromosomal locations and possess varying genetic structures, also display similar functional disparities remains limited. Further analysis is warranted to elucidate this matter. Utilizing amphioxus from QingDao as our material, we successfully cloned these two homologous sequences, naming them *cyp19-like1* and *cyp19-like2*. 

Given the presence of amphioxus *cyp19-like1* and *cyp19-like2*, which are also situated on different chromosomes and display structural and sequence variations, it raises intriguing questions regarding their specific functions in amphioxus and whether both genes play a role in sex differentiation. *cyp19-like1* has 10 exons, while the other sequences possess 9 exons, which aligns with the basic structure of vertebrate *cyp19* having 9 exons [23,24]. Amphioxus and zebrafish share a conserved cytochrome p450 domain, indicating a common ancestral origin. Notably, zebrafish Cyp19a1a lacks a transmembrane region, whereas zebrafish Cyp19a1b, amphioxus Cyp19-like1, and amphioxus Cyp19-like2 all possess a transmembrane region at their N-termini. This feature suggests potential differences in their subcellular localization and function.

The question at hand concerns the functional relationship between the two *cyp19-like* in amphioxus and the *cyp19a1a* and *cyp19a1b* in fish, particularly in the context of their evolutionary history. Zhang and colleagues’ observations [44] about the strong synteny and sequence conservation of vertebrate *cyp19a1* during evolution provide a valuable baseline for comparison. Despite this conservation, amphioxus Cyp19 (ABA47317.1) does not share synteny with vertebrate *cyp19a1*, suggesting that the direct evolutionary link between these two genes is tenuous. In their opinion, it appears that the *cyp19a1* in bony fish may have originated from an ancestor that evolved alongside amphioxus, rather than directly descending from it. Alternatively, significant chromosomal rearrangements in the region surrounding the *cyp19a1* locus could have occurred in the basal vertebrate ancestor closely related to amphioxus [44]. Our findings, however, introduce a new element to this discussion. Analysis indicates that the two *cyp19-like* sequences in amphioxus do not exhibit a one-to-one correspondence with *cyp19a1a* and *cyp19a1b* in zebrafish. Amphioxus *cyp19-like1* may represent the primitive form preceding the evolution of vertebrate *CYP19* genes, while *cyp19-like2* could be the result of gene duplication specific to amphioxus.

Our analysis indicates that amphioxus *cyp19-like* exhibit some synteny with vertebrate *cyp19a1*. Yet, *cyp19-like2* exhibits stronger synteny with humans, mice, and zebrafish. This suggests that *cyp19-like2* may have a closer evolutionary affinity with these vertebrate genes. In addition, phylogenetic analysis provides valuable insights into the evolutionary history and potential functions of genes. In the context of *cyp19*, the finding that *cyp19-like1* represents a more primitive and ancestral form suggests that it played a crucial role in the early stages of vertebrate evolution. The proposed origin of *cyp19-like2* from genome duplication within amphioxus itself highlights the dynamic nature of genome evolution and the potential for novel functions to arise from such duplications.

In bony fish, the observed differences in tissue expression level [47,48], affinities for substrates [49,50], and inducibility by estrogens and xenoestrogens [48,51,52,53] between *cyp19a1a* and *cyp19a1b* are consistent with the idea that duplicated genes can diverge in function. Similar observations in pigs further support this notion [54,55]. Given these findings, it is intriguing to explore whether the two amphioxus *cyp19-like* exhibit differences in expression level, particularly in male and female gonads. Our study using real-time PCR and ISH techniques has revealed an interesting level of *cyp19-like* expression in amphioxus. The high expression of *cyp19-like1* in the gills, followed by the ovaries, suggests a role for this gene in reproductive and other functions. In contrast, the predominant expression of *cyp19-like2* in the ovaries is consistent with its putative role in sex differentiation. We investigated whether there are differences in *cyp19-like* expression in the gills of males and females using real-time PCR and found no sex-specific differences in expression in the gills. The absence of sex-specific differences in *cyp19-like1* expression in the gills is noteworthy, indicating that this gene may play a more general role in amphioxus physiology. However, the expression level of *cyp19-like* in the heads of adult amphioxus remains enigmatic and deserves further investigation.

The examination of *cyp19* expression during gonad development in amphioxus provides further insights into the potential roles of these genes in sex differentiation and reproduction. The high expression of both *cyp19-like* at the onset of gonad development suggests their involvement in the initial stages of gonad formation. However, the subsequent downregulation of expression as development progresses might indicate that these genes are not essential for maintaining gonad function but rather play a crucial role in initiating gonad development. Immunohistochemical staining also confirmed that the proteins encoded by these two genes were most abundant in the early stages of gonad development, with subsequent downregulation. Overall, the expression of *cyp19-like2* and its encoded protein was more prominent in the gonad primordium. Aromatase proteins are monomeric and anchored within the endoplasmic reticulum via a transmembrane domain at their amino-terminal end [56,57]. We investigated the subcellular localization of Cyp19 and found that the protein is likely localized primarily to microsomal membrane systems such as mitochondrial and endoplasmic reticulum membranes, consistent with its expression level in vertebrate cells.

Future studies should aim to further investigate the molecular mechanisms underlying the observed differences in gene expression levels during gonad development and elucidate the precise functions of *cyp19-like1* and *cyp19-like2* in amphioxus, including their potential roles in sex differentiation, reproduction, and other physiological processes.

## 5. Conclusions

We comprehensively examined the expression of two *cyp19-like* genes in amphioxus using techniques such as real-time PCR, ISH, and IHC. This study clarifies the Cyp19 expression level during amphioxus gonad development and highlights its importance in vertebrate reproduction evolution. Differences in gene expression and protein localization of *cyp19-like1* and *cyp19-like2* offer insights into sex determination and gonad development mechanisms. However, further research is needed to fully understand these complex mechanisms.

## Figures and Tables

**Figure 1 animals-14-01140-f001:**
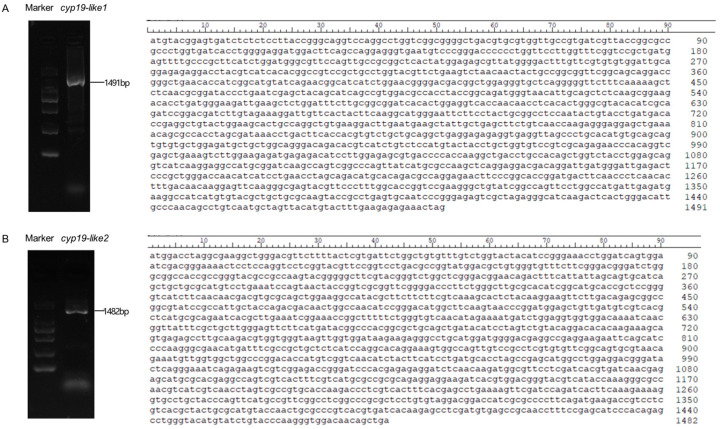
Complete ORF nucleotide sequences of two *cyp19-like* fragments in amphioxus. (**A**): The *cyp19-like1* ORF in amphioxus, with a size of 1491bp. (**B**): The *cyp19-like2* ORF in amphioxus, with a size of 1482bp.

**Figure 2 animals-14-01140-f002:**
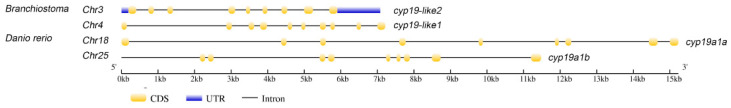
Comparison of the gene structures of *cyp19a1a*, *cyp19a1b*, and *aromatase-like* genes in zebrafish and Florida amphioxus.

**Figure 3 animals-14-01140-f003:**
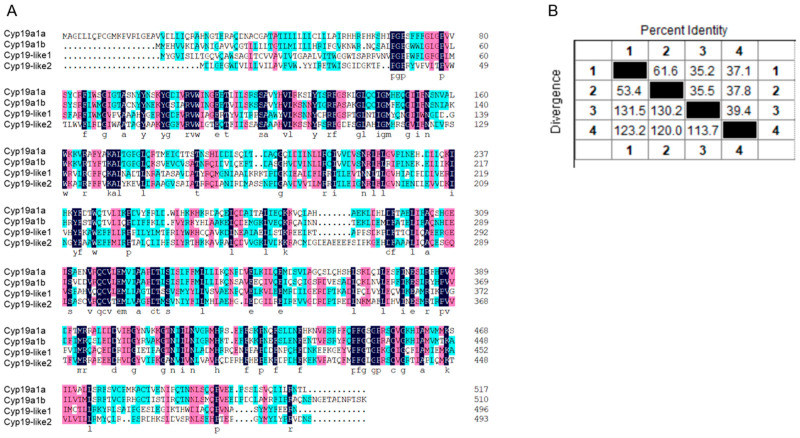
Alignment of amino acid sequences of *cyp19* in zebrafish and amphioxus. (**A**): Amino acid sequence alignment; (**B**): sequence similarity comparison. 1 represents *cyp19a1a*, 2 represents *cyp19a1b*, 3 represents *cyp19-like1*, and 4 represents *cyp19-like2*.

**Figure 4 animals-14-01140-f004:**
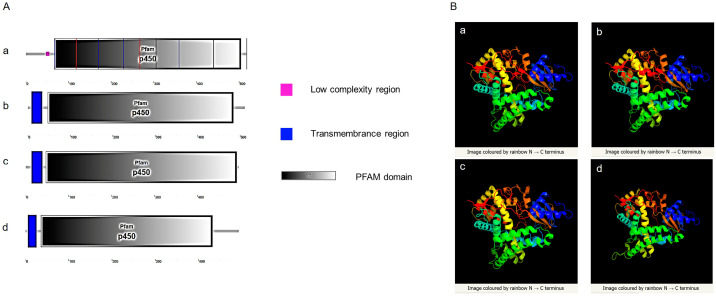
Prediction of Cyp19 structures in zebrafish and amphioxus. (**A**): SMART prediction of domains in zebrafish Cyp19a1a (**a**), Cyp19a1b (**b**), and amphioxus Cyp19-like1 (**c**), Cyp19-like2 (**d**); (**B**): PHYRE2 prediction of three-dimensional structures in zebrafish Cyp19a1a (**a**), Cyp19a1b (**b**), and amphioxus Cyp19-like1 (**c**), Cyp19-like2 (**d**).

**Figure 5 animals-14-01140-f005:**
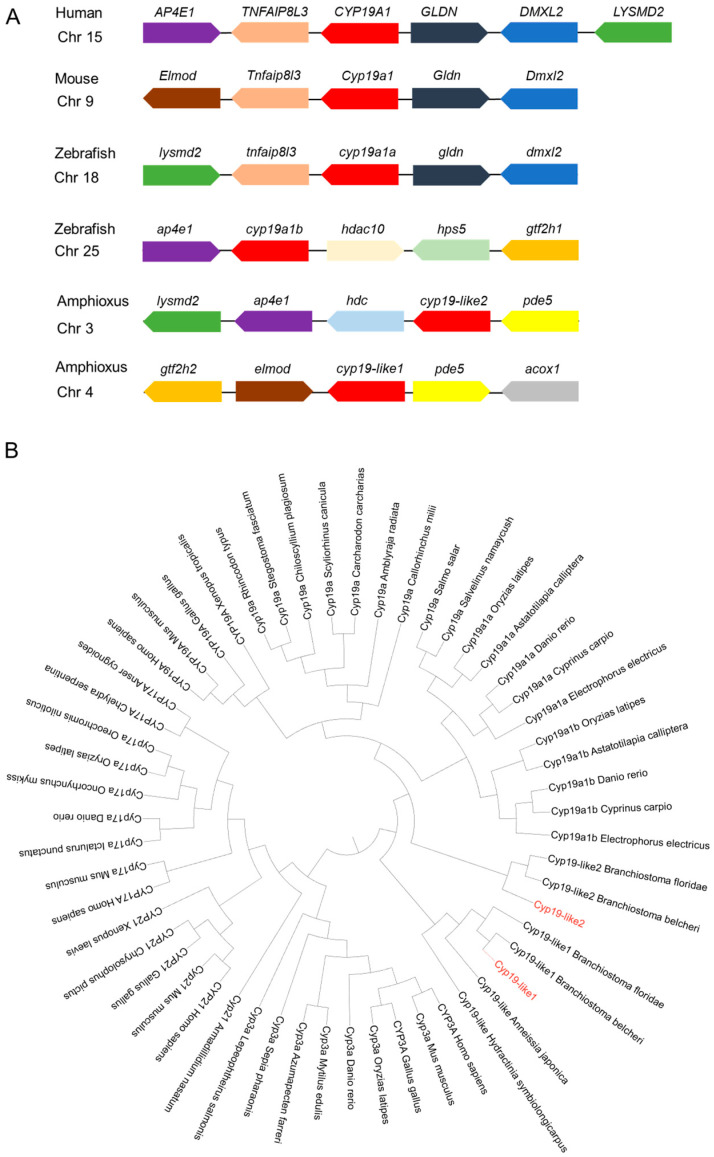
Analysis of synteny and phylogeny of Cyp superfamily across different species. (**A**): Location maps of Cyp19 on amphioxus and other vertebrate chromosomes. Boxes represent genes, and the direction of the boxes indicates the orientation of the genes. (**B**): Phylogenetic analysis of Cyp superfamily across different species. Sequence sources: see Appendix A.

**Figure 6 animals-14-01140-f006:**
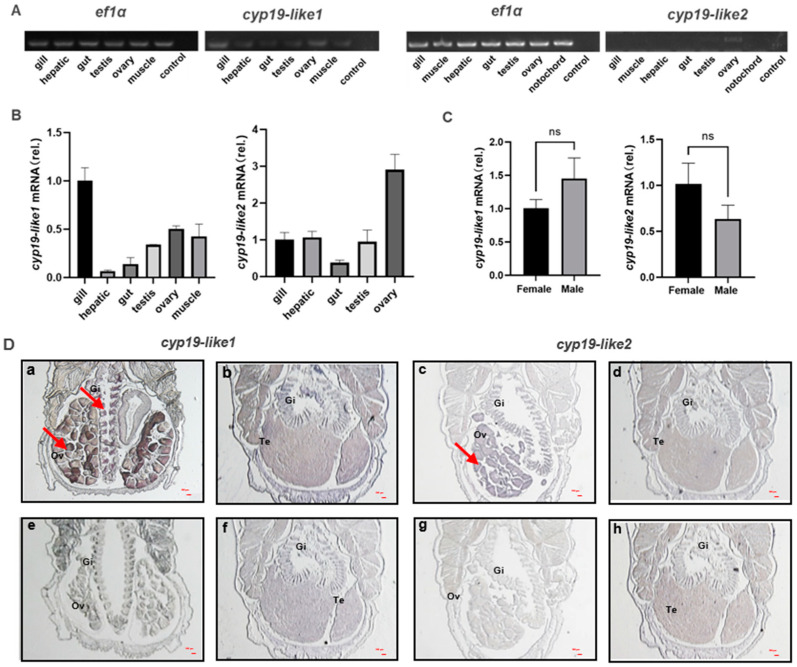
Tissue expression and localization of *cyp19-like* in amphioxus. (**A**): Semi-quantitative PCR detection of *cyp19-like* gene expression in different tissues of amphioxus. (**B**): Real-time PCR detection of *cyp19-like* gene expression in different tissues of amphioxus, including gill, hepatic, gut, testis, ovary, muscle, and notochord; (**C**): expression of *cyp19-like* in the gills of male and female amphioxus. ns indicates no significant difference. (**D**): ISH results of *cyp19-like* on paraffin sections of amphioxus. (**a**–**d**) show antisense probe hybridization results; (**e**–**h**) show sense probe hybridization results (negative control). Gi: gill; Ov: ovary; Te: testis. Arrows indicate positive signals. Scale bar: 100 μm.

**Figure 7 animals-14-01140-f007:**
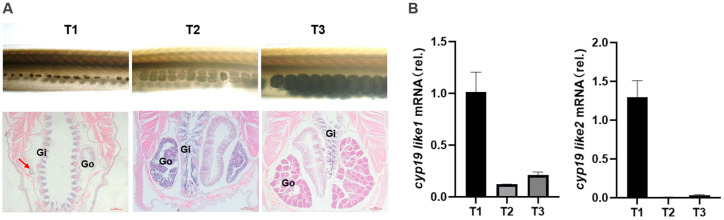
Expression of *cyp19-like* in amphioxus at different developmental stages. (**A**): Tissue morphology and HE staining results at different stages of gonad development in amphioxus. Gi: gill; Go: gonad; arrows indicate gonadal primordium. (**B**): Real-time PCR detection of *cyp19-like* expression at different stages of gonad development in amphioxus. T1, T2, and T3 represent the primordial, developing, and mature gonad stages, respectively.

**Figure 8 animals-14-01140-f008:**
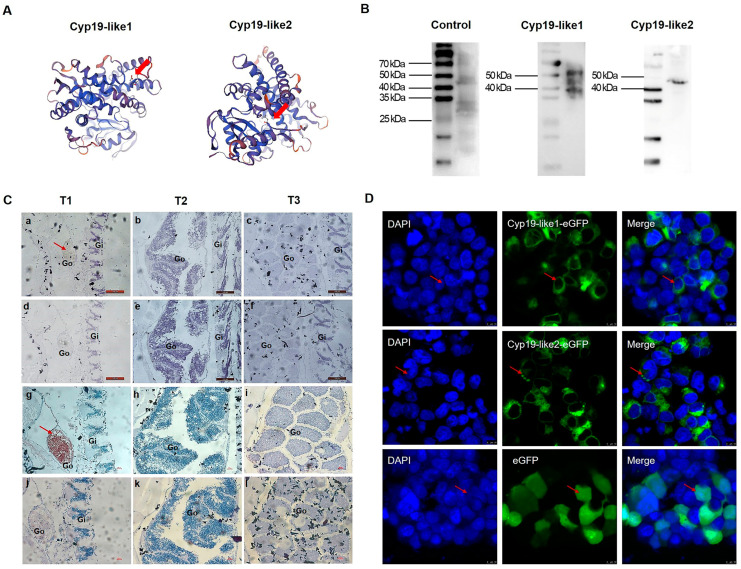
Antibody preparation and protein expression of amphioxus Cyp19-like. (**A**): Designated sites for immunogenic peptides which are indicated with arrows. (**B**): Western blot results of Cyp19-like in amphioxus. (**C**): Immunohistochemical results of Cyp19-like in amphioxus. T1, T2, and T3 represent the primordial, developing, and mature gonad stages, respectively. a–f show Cyp19-like1 immunohistochemical results, while g–l show Cyp19-like2 immunohistochemical results. a–c, g–i are experimental groups, while d–f, j–l are control groups. Gi: gill; Go: gonad; arrows indicate positive signals. Scale bar: 100 μm. (**D**): Subcellular localization of amphioxus Cyp19-like in HEK cells using LSCM. The expression of Cyp19-like is depicted in green; nuclei were counterstained with DAPI (blue). Red arrow indicating positive signal. Scale bar: 10 μm.

**Table 1 animals-14-01140-t001:** The sequences of primers and peptides used in this study.

Experiment	Description	Sequences (5′–3′)
PCR	P1(*cyp19-like1* F)	ATGTACGGAGTGATCTCTCTCCTTA
	P2(*cyp19-like1* R)	CTAGTTTCTCTCTTCAAAGTACATG
	P3(*cyp19-like2* F)	ATGGACCTAGGCGAAGGCTGGGACG
	P4(*cyp19-like2* R)	TCAGCTGTTGTCCACCCTTGGGTAC
Real-time PCR	P5(*ef1α* F)	TGCTGATTGTGGCTGCTGGTACTG
	P6(ef1α R)	GGTGTAGGCCAGCAGGGCGTG
	P7(*cyp19-like1* F)	GCTCAGGAGGACGACAGGATTG
	P8(*cyp19-like1* R)	GCAGCAGCGTACACATGATGG
	P9(*cyp19-like2* F)	TTCGCCGCTGCTCTCATCCA
	P10(*cyp19-like2* R)	CGGTCTCCGACGACTTCTCTGA
In situ hybridization	P11(*cyp19-like1* F)	GCGTGGTCGCCGTTGTCGTT
	P12(*cyp19-like1* R)	CGCCGCAAGAAATCCAGAGCT
	P13(*cyp19-like2* F)	GTGTATCCGCCATTGCTACC
	P14(*cyp19-like2* R)	TCTCCGACGACTTCTCTGATT
Polyclonal antibody	Cyp19-like1	N′-CREELKTAPPSDKPD-C′
preparation (mouse)	Cyp19-like2	N′-CPSRDHKSLDVSRNL-C′
Subcellular Localization	P15(*cyp19-like1* F)	gcacagtggcggccgctcgagATGTACGGAGTGATCTCTCTCCTTACC
	P16(*cyp19-like1* R)	gctcaccattctagactcgagGTTTCTCTCTTCAAAGTACATGTAACTAGC
	P17(*cyp19-like2* F)	gcacagtggcggccgctcgagATGGACCTAGGCGAAGGCTG
	P18(*cyp19-like2* R)	gctcaccattctagactcgagGCTGTTGTCCACCCTTGGG

## Data Availability

All relevant data are available from the authors upon request and the corresponding author will be responsible for replying to the request.

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
