# Peer review of "Identification, Expression and Evolutional Analysis of Two *cyp19-like* Genes in Amphioxus"

_animals, 2024, doi:10.3390/ani14081140_

Round 1
Reviewer 1 Report
Comments and Suggestions for Authors
In this study, authors delve into the evolution of sex determination and differentiation in animals, with a special focus on the cephalochordate amphioxus. The study identifies two cyp19 homologous genes in amphioxus, cyp19-like1, and cyp19-like2, reporting the evolution, synteny, and expression patterns of these genes in male and female gonads.
Although the experimental setup is properly designed the manuscript missing the appropriate description of the fundamental part as Material and Methods, as well as the need to better argue and describe different parts of this manuscript.
Major:
Line 46-47: the authors should better introduce the sex determination in invertebrates maybe adding some examples and reporting references.
In the Materials and Methods section, the precise protocol and procedures employed should be reported, indicating all the useful information for repeating the experiments. In here general information on technique and general protocols were provided. This section has to be completed and rewritten.
Through the text the figures should be named as “Figure 1, panel A” and not Figure A.
Line 205-209: The sentences show several inaccuracies. 5’RACE is not considered a technology and is not suitable for cloning. Additionally, the authors state in the introduction that they found two cyp19-like genes on separate chromosomes, but fail to explain what cyp19-like1 and cyp19-like2 are. In the legend of Figure 1, they are defined as “fragments” inducing the reader to interpret them as different portions of the same gene.
Line 223: The position of the genes on different chromosomes does not justify a putative different function.
Line 265: the synteny between cyp19-like2 with humans, mice, and zebrafish, it does not demonstrate any role in sex determination, only a putative common evolutive history.
Line 325: the expression level of cyp19-like1 and cyp19-like2 T0 should be evaluated, or if it is evaluated and used as a “control stage” it should be expressly stated in the text. Moreover, a histological section (at T0) of the area where the gonads will form could be useful.
Line 355: Any Antibody preparation was described in the text. It is not clear if the antibody used for the immunostaining procedure was a commercial one or specifically designed on amphioxus epitoph.
Minor:
Through the text, please uniform the style of the gene names and their orthologs/paralogs. I suggest to follow Z-fin Nomenclature rules. https://zfin.atlassian.net/wiki/spaces/general/pages/1818394635/ZFIN+Zebrafish+Nomenclature+Conventions
For example, Cyp19a1a (Line 69) and Cyp19a1b (Line 71) should be cyp19a1a and cyp19a1b, as in Line 68.
The quality of in situ hybridization images does not allow for verification of the gene expression analyses.
Line 33: The keyword Cyp19 should be in Italic style
Line 44: Before “since” a space is necessary
Line 66: I did not get the sense of the expression “ the Actinopterygii of Osteichthyes”, please reformulate e
Line 289-291: In situ have to be in Italic
Line 321: Expression pattern should be changed in expression level.
Line 323: define the beginning time with the T scheme (T to T3)
Comments on the Quality of English LanguageModerate editing of English language are required
Reviewer 2 Report
Comments and Suggestions for Authors
The manuscript entitled „Identification, expression and evolutional analysis of two 2cyp19-like genes in amphioxus“ deals with important evolutionary issues. The authors try to provide valuable insight into the functions of conserved genes and pathways in genetic sex determination. The obtained results represent valuable data on expression patterns during amphioxus gonad development. The comparation to already known data on 2cyp19-like genes was done in appropriate way in correspondence to used methods. The topic is particularly relevant because it confirmed the well-known and important role of amphioxus in vertebrate evolution and it brings novelty from the evolutionary point of view.
The Materials and Methods section should be written in past tense and described in detail provided with the names of solutions and equipment used in the experimental work. If these methods have been already used then appropriate references should be cited. Almost every figure has an A, B or C part which requires the number also when describing and explaining in the text (Figure 3A and Figure 3B instead Figure A and Figure B).
In section 3.2.4. Synteny and Evolutionary Analysis ofCyp19 a few conclusions were stated, it should be in the discussion section.
Figure 8C and 8D – certain parts of tissue should be labelled, significant findings as well, as it was done on Figure 7A.
All figures should have an appropriate explanation of the abbreviations.
Line 386 - a year is missing in the reference.
Reviewer 3 Report
Comments and Suggestions for Authors
The manuscript entitled " Identification, expression and evolutional analysis of two 2
cyp19-like genes in amphioxus" is a good attempt. In this study, cyp19-like1 and cyp19- like 2 were identified in amphioxus, providing important insights into the mechanisms shaping sex differentiation sex determination in amphioxus. The article is interesting for reader, but it should be improved as follows:
1. line20, line63: “Cytochrome P450 Family 19 Subfamily A member 1” does not need to be written in italics.
2. For example: line69, line71, line73, line380-381, line249, etc.: We found that things like Cyp19a1a ,Cyp19a1b, Cyp19-like1, and Cyp19-like2 h are not italicised, and I have found that genes denoted in the text but not italicised are still present, and I suggest that the authors double-check the whole thing, I suggest the authors to check the whole paper carefully and make corrections.
3. In the Introduction section, it is suggested that the research progress of amphioxus in terms of gender could be expanded and supplemented as appropriate.
4. In 2.2, please add a description of the method and reagents used to extract the RNA and how the quality of the RNA is determined
5. In 2.3, the description of the gene cloning method is written too briefly, and it is recommended that more information be added, including the reagents used and PCR amplification conditions.
6. In 2.5, it is recommended to supplement the reagents, reaction conditions, etc. used during the experiments for real-time fluorescence quantification。
7. why did you chose P6(actin ) as the reference gene?
8. Figure 1: Lines 207-208: It is recommended that (Figure A), (Figure B) be changed to (Figure 1A). (Figure 1B), Similarly, Figures 2-8 are proposed to be modified accordingly.
9. In table 1,Should P6 (actin F) be changed to P6 (actin R)?
10. In images 8-B and 8-D, Cyp19-like1, Cyp19-like2 are not labelled in italics in the figure
Comments on the Quality of English LanguageNo.
Round 2
Reviewer 1 Report
Comments and Suggestions for Authors
The authors have addressed all of my requests thoroughly, resulting in a comprehensive revision of the previously missing Material and Methods section. After these revisions, I propose the manuscript for publication without any additional modifications.
Line 49: Daphnia magna shoud be italic
Reviewer 3 Report
Comments and Suggestions for Authors
The answers are ok.